# A TSENet Model for Predicting Cellular Network Traffic

**DOI:** 10.3390/s24061713

**Published:** 2024-03-07

**Authors:** Jianbin Wang, Lei Shen, Weiming Fan

**Affiliations:** 1Ocean College, Zhejiang University, Zhoushan 316021, China; zjtelecom@zju.edu.cn; 2Wireless Network Center, China Telecom Corporation Limited Zhejiang Branch, Hangzhou 310005, China; 3Communication Engineering College, Hangzhou Dianzi University, Hangzhou 310018, China; 4With the Innovation Center for Smart Medical Technologies & Devices, Binjiang Institute, Zhejiang University, Hangzhou 310053, China; weiming@stu.sylu.edu.cn

**Keywords:** cellular network, WSNs, TSENet, traffic prediction, self-attention aggregation

## Abstract

Wireless sensor networks (WSNs) are gaining traction in the realm of network communication, renowned for their adaptability, configuration, and flexibility. The forthcoming network traffic within WSNs can be forecasted through temporal sequence models. In this correspondence, we present a method (TSENet) that can accurately predict the traffic in the cellular network. TSENet is composed of transformers and self-attention network. We have designed a temporal transformer module specifically for extracting temporal features. This module accomplishes this by modeling the traffic flow within each grid of the communication network at both near-term and periodical intervals. Simultaneously, we amalgamate the spatial features of each grid with information from its correlated grids, generating spatial predictions within the spatial transformer. Furthermore, we employ self-attention aggregation to capture dependencies between external factor features and cellular data features. Empirical assessments performed on a genuine cellular traffic dataset offer compelling evidence substantiating the efficacy of TSENet.

## 1. Introduction

In recent years, the swift progress in distributed computing and micro-electromechanical systems (MEMS) has propelled wireless sensor networks into numerous domains, encompassing military [1], intelligent architecture [2], and industrial sectors [3,4]. To observe diverse phenomena like network traffic, sensors are equipped with sensing, computing, communication, and additional functionalities.

With the continuous development of wireless communication technologies, higher requirements are being imposed on wireless communication networks. Among these requirements, accurately predicting the future change trend of key performance indicators of wireless cells holds crucial significance. It profoundly impacts the network planning, network optimization, and network energy-saving practices of mobile communication operators.

Predicting the change trend of the key performance index of the wireless cell is very important for the mobile communication operator, which can help the operator to plan the network. By accurately predicting future KPI changes, operators can properly plan network resource allocation, ensure stable network connection during peak hours, and avoid congestion and service quality deterioration. By learning the change trends of performance counters, operators can adjust network parameters and configurations in a timely manner to optimize network performance and provide better user experience. For example, according to the prediction result, the operator may decide whether to increase the number of base stations, adjust the power allocation, or change the channel allocation policy, thereby increasing the network capacity and coverage area.

In recent studies, Yu et al. [5] tackled the challenge of cellular data prediction with STEP, a spatiotemporal fine-grained mechanism utilizing GCGRN. The deep graph convolution network outperformed existing methods, offering efficiency and reduced resource consumption in cellular networks. Santos et al. [6] evaluated DL and ML models for short-term mobile Internet traffic prediction. LSTM exhibited excellent performance across diverse clusters in Milan. Hardegen et al. [7] presented a pipeline using deep neural networks (DNNs) to predict information of real-world cellular data flows. Unlike traditional binary classifications, their system quantizer predicted flow features into three classes.They emphasized the potential for flow-based routing scenarios. Furthermore, Dodan et al. [8] proposed an efficient Internet traffic prediction system using optimized recurrent neural networks (RNNs). They addressed the challenges in large-scale networks, emphasizing the need for scalable solutions to anticipate traffic fluctuations, avoid congestion, and optimize resource allocation. Experimental outcomes showcased the enhanced efficacy of their RNN-based methodology compared to conventional approaches such as ARIMA and AdaBoost regressor.

In the realm of deep learning methodologies, both convolutional neural networks (CNNs) [9] and recurrent neural networks (RNNs) [10] stand as foundational structures for modeling temporal information. Cellular communication data exhibit temporal periodicity in their spatial distribution, bearing distinct characteristics [11,12]. Consequently, contemporary deep learning models aim to integrate temporal and spatial information in cellular data. Zhang et al. [13] came up with STDenseNet, a convolutional neural network approach for cellular data prediction. In RNN models, elements in the input sequence are processed sequentially, leading to inefficiencies in handling lengthy sequences. Conversely, methodologies primarily reliant on CNNs may fall short in fully capturing spatial dependencies between adjacent features. Additionally, approaches leveraging RNNs for extracting prolonged temporal relationships exhibit inherent weaknesses. These constraints may result in an insufficient description of continuous and temporal features.

In response to these challenges, we present a model named TSENet, founded on the principles of transformers and self-attention. Initially introduced in 2017 within the realm of natural language processing, the transformer architecture boasts a parallel encoder–decoder architecture, later extended to various domains. Primarily, we harness and tailor its robust sequential modeling capabilities. Concretely, we devise a time-sequence module (TSM) and a spatial-sequence module (SSM), adept at simultaneously and accurately extracting temporal and spatial features within short time intervals. Subsequently, we employ self-attention mechanisms to amalgamate the model’s prowess in capturing dependencies between external factors and cellular traffic characteristics. Our comprehensive experiments conducted on a bespoke dataset aim to substantiate the efficacy of our approach. Subsequent analyses and discussions delve into a nuanced exposition of the TSENet model.

In summary, the primary contributions of this study can be delineated as follows: (1) We pioneer the introduction of a spatiotemporal traffic forecasting model, christened TSENet, which harmoniously integrates external elements. This model, predicated on harnessing spatiotemporal features, takes into account the nuanced characteristics of external factors. (2) By employing attention mechanisms, TSENet adeptly captures the multifaceted global spatiotemporal correlations within cellular traffic networks. Notably, TSENet extracts both enduring and fleeting temporal features, culminating in the establishment of a spatial nexus model among nodes. (3) Through the incorporation of self-attention mechanisms, TSENet effectively excavates the intricate interdependencies between external factors and cellular traffic, thereby further elevating the predictive precision of TSENet.

## 2. Related Work

### 2.1. Wireless Sensor Networks

Network sensing focuses on capturing the dynamic state and performance metrics of the cellular network itself, such as signal strength, latency, and packet loss. On the other hand, environmental sensing involves monitoring external factors like weather conditions, interference, and population density. The features extracted from both network and environmental sensing contribute to the comprehensive set of parameters used for accurate network traffic estimations. Wireless sensor networks epitomize the avant-garde paradigm in ubiquitous computing, embodying the fusion of sophisticated sensor technologies and wireless communication. The structural backbone of these networks comprises diminutive yet potent sensor nodes. Communication among these sensors unfolds through wireless protocols, fostering seamless interconnectivity between nodes [14]. This mirrors the traditional single-receiver WSN (refer to Figure 1, left segment). However, this singular receiver scenario lacks scalability. Due to the substantial number of nodes, the collector accumulates more data, impeding the network’s scalability when it reaches capacity. This is a frequent scenario, especially with multiple receivers in a network (depicted on the right side of Figure 1) [15]. This decentralized approach exhibits advantages in terms of scalability, fault tolerance, and load distribution. In practical applications, wireless sensors serve diverse purposes in the realm of cellular traffic monitoring. Krishnasamy et al. [16] provided an overview of wireless sensor networks, comprising multiple sensors with sensing and communication functionalities. These sensors both receive and transmit data to designated nodes. The core purpose of a sensor node is to monitor environmental conditions, process the acquired data, and then transmit it to an analysis center. Sensor deployment locations are typically nonuniformly designed, often installed randomly in irregularly shaped specific locations based on their transmission range. Hence, it is important that the deployed algorithms can be adapted to each different geographical area. In essence, deploying wireless sensors facilitates the analysis of traffic patterns, detection of anomalous behavior, optimization of cellular networks, and predictive modeling.

### 2.2. Network Traffic Prediction

Network traffic prediction has been the subject of prolonged inquiry. ARIMA [17] and Holt–Winters (HW) [18] are both univariate time series models. ARIMA leverages autoregressive, difference, and moving average components, with the component order determined by Akaike or Bayesian information criteria. Kim et al. [19] introduced the integer-valued generalized autoregressive conditional heteroscedasticity (INGARCH) as a nonlinear model for network traffic prediction to study traffic arrival processes in various network topologies, including IoT and VANETs. The results suggested that INGARCH was a suitable and tractable model for network traffic prediction, performing competitively or superiorly to other statistical models, particularly in data-insufficient scenarios compared to deep neural networks. In contrast to statistical models, ST-GPKL models often show superior capabilities [20]. Xia et al. [21] applied RF and LightGBM to the network traffic prediction model. Zhang et al. [22] employed a model based on Gaussian functions and a weighted EM method to optimize UAV deployment for reduced power consumption in downlink transmission and mobility. Traffic prediction in machine learning often relies on supervised neural networks like MLP and RNN in time series forecasting (TSF) approaches, demonstrating high accuracy with limited complexity [23]. Non-TSF methods explore inferring traffic volumes from flow count and packet header fields, showing lower but acceptable error rates, motivating further research in ensemble learning for enhanced accuracy and automation of network management activities. Related work in machine learning traffic prediction involves addressing challenges related to ground truth (GT), emphasizing the scarcity of labeled data in dynamic environments like network traffic. Strategies such as PCA and active Learning are explored to overcome this limitation [24]. In recent years, neural networks have emerged as potent tools, attaining significant success across various prediction domains, notably in traffic prediction challenges. Basic yet pivotal, the feed-forward neural network serves as the cornerstone for cellular traffic prediction, incorporating hidden layers and diverse activation functions like tanh, sigmoid, and ReLU to enhance learning capabilities. However, for spatial–temporal prediction challenges, a flattening calculation is required, potentially leading to a lack of spatial information and a decline in prediction performance.

### 2.3. Transformer

In recent years, endeavors rooted in the transformative power of transformers have burgeoned across domains such as computer vision, signal processing [25], and bioinformatics [26], yielding superlative outcomes in these realms. An array of studies has been conducted to unravel the potential of transformer networks in forecasting network traffic, demonstrating their superiority over conventional machine learning methodologies. Wen et al. [27] proffered a transformer-based model meticulously designed for the prediction of short-term traffic congestion. This model, when juxtaposed against traditional machine learning methods like LSTM [28], exhibited a pronounced prowess. Qi et al. [29] expounded upon the performer model, an innovative transformer-based predictive methodology for clustering voluminous time series data, showcasing conspicuous efficacy in comparison to conventional methods such as random forest and SVM. Chuanting Zhang et al. [30] presented a groundbreaking framework for wireless traffic prediction, termed federated dual attention (FedDA). FedDA facilitates collaborative training among multiple edge clients to cultivate a high-fidelity predictive model, eclipsing state-of-the-art methods with substantial enhancements in average mean squared error performance. Rao et al. [31], leveraging deep learning techniques, undertook an exhaustive exploration of challenges and recent strides in cell load balancing. Feng et al. [32] propounded an end-to-end framework christened as DeepTP, meticulously designed for predicting traffic demands from mobile network data through the prism of deep learning techniques. DeepTP not only demonstrated a superiority over other cutting-edge traffic prediction models by exceeding them but also adeptly captured the spatiotemporal characteristics and external factors’ influence for a more precise predictive outcome.

Despite achieving commendable milestones in internet traffic prediction, further exploration is requisite regarding the application of cellular traffic data models based on wireless sensor networks. Diverging from prior endeavors, the utilization of self-attention mechanisms has been employed to congregate the extraction of external factors and dependencies in network traffic features, thereby judiciously harnessing external influences. Moreover, we employ the transformer architecture to capture temporal and spatial information within concise intervals. Through the efficacy of multi-head self-attention mechanisms, we adeptly discern spatial relationships effectively interlaced between grids. This novel structure holds promise in mitigating, to some extent, the limitations inherent in existing methodologies.

## 3. Method

In this paper, we use the self-attention mechanism to aggregate external factors and dependencies in network traffic features and use a transformer to capture temporal and spatial information in concise intervals. Through the effectiveness of multi-head self-attention mechanism, we can skillfully identify effective interlaced spatial relationships between grids. This section describes the TSENet framework, including its spatial-sequence module, time-sequence module, external factor module, and fusion output module.

### 3.1. The Proposed TSENet Framework

This article introduces TSENet, which comprises four primary modules: the time-sequence module, spatial-sequence module, external-factor module, and fusion output module, as illustrated in Figure 2. The time-sequence module and spatial-sequence module are designed based on the fundamental structures of the transformer. The TSM extracts recent temporal trends in cellular traffic $M_{c}\left(t\right)$ and periodic features in $M_{p}\left(t\right)$. The calculation block fuses them into temporal features. Concerning spatial feature modeling, the multi-head attention mechanism in a series of networks can effectively capture spatial relationships between network features. Real-time spatial features are captured in the SSM. The EFM is designed using a self-attention structure to extract external factor features. These three types of features complement each other, resulting in the generation of the final prediction through the FOM. The transformer enables elements of the input sequence to contribute to the result, incorporating target sequence prediction values in the decoding layer, thus achieving more effective predictions using additional information.

### 3.2. Spatial-Sequence Module

In the SSM, the cellular traffic data of the pertinent network serve as elements within the input sequence. We use a multi-head attention structure in order to engender spatial predictions. It processes information from each network and its related networks. For each network, analyzing its relationship with all other networks yields a considerably extensive grid sequence. Indeed, only a subset of networks significantly influences the traffic states of others. Hence, we select the *L* networks most correlated with the current network (inclusive of the network itself). The chosen traffic data from $M_{c}\left(t\right)$ is linked to a new sequence, $M_{s}\left(t\right)\in R^{H\times L\times J}$, serving as input to the spatial-sequence block, wherein *L* is the length of the spatial array, and *J* denotes the dimension of each array.

(1)Grid arrangement: We sort each grid according to its correlation with other grids and select the top *L* largest grids. The value of *L* can be chosen based on experimental results. We employ the Pearson correlation coefficient to assess the correlation. The correlation between two grids A and B with equal traffic sequence lengths is calculated by the covariance of the variables and the product of their standard deviations:
(1)$$R_{A,B}=\frac{cov(A,B)}{\sigma A\sigma B}.$$We concatenate the incoming data $A_{in}\left(t\right)\in R^{H\times J}$ and outgoing data $A_{out}\left(t\right)\in R^{H\times J}$ to form a matrix of dimensions H×2J. Subsequently, the correlation matrix $P\left(t\right)\in R^{H\times H}$ is computed according to Equation. $P_{i,j}$ signifies the relevance between the *i* and *j* grids at the *t* interval.(2)Spatial sequence: As illustrated in Figure 2, the source sequence and the initial sequence serve as the principal inputs to the essential transformer. In natural language processing (NLP), the elements of the sequence involve more informative word embeddings, but the array of cellular traffic data results in the scarcity of information. We choose the two complementary input arrays to ensure the predictive performance of the transformer. The proximity matrix $M_{c}\left(t\right)$ signifies the cellular communication states close to the target interval. The $M_{s}\left(t\right)$ designed by $M_{c}\left(t\right)$ encompasses real-time spatial features for the sequence with a length of *L*, serving as the source array, and $M_{c}\left(t\right)\in R^{L\times 1\times J}$ is taken as the initial array with a length of 1, yielding the output of the SSM, $Y_{spatial}\left(t\right)\in R^{L\times J}$. We omit positional encoding because there is no chronological order between spatial networks.

### 3.3. Time-Sequence Module

As depicted in Figure 2, the TSM employs two temporal transformers to individually acquire features from near-term and periodic continuous time. The calculation block utilizes a ⊕ operation and an SSM to aggregate these two features into a predictive time sequence.

(1)Closeness sequence: To harness additional information from the input data, $M_{c}\left(t\right)$ is extended as the source sequence of the proximity sequence. During the enhancement procedure, we use the correlation matrix $P_{t}$ to select the topmost U correlated networks for each grid. We combine the selected data and the grid data and create a scalar with dimension L×J×(1+U), where *U* is a parameter determined through experimentation. This process enhances both the temporal features within the sequence elements and introduces a few space information in the original sequence, thereby enhancing the accuracy of predictions.In the closeness sequence, the averaging of the second dimension of $M_{p}\left(t\right)\in R^{L\times G\times J}$ yields an initial sequence of dimensions L×J×1. We employ these averaged historical data as indicators for the traffic data levels in future time intervals. The closeness sequence produces the proximity time prediction, denoted as $Y_{c}\left(t\right)\in R^{L\times J}$.(2)Periodic sequence: The periodic sequence is akin to the proximity sequence but lacks critical information. We employ $M_{p}\left(t\right)\in R^{L\times G\times J}$ as the original sequence, with $M_{c}\left(t\right)\in R^{L\times 1\times J}$ serving as the initial sequence for the temporal sequence, thereby utilizing the compact time information included in $M_{c}\left(t\right)$ to complement the period sequence. The outcome of the periodic time prediction is expressed as $Y_{p}\left(t\right)\in R^{L\times J}$.

The time-sequence module mixes $Y_{c}\left(t\right)$ and $Y_{p}\left(t\right)$ up into the final temporal prediction $Y_{time}\left(t\right)\in R^{L\times J}$.

### 3.4. External-Factor Module

The quantity of base stations (BSs), points of interest (POIs), and social activities within a specific region serves as a direct indicator of users’ capacity to request telecommunication services. By leveraging the geographical location details of each community, we can derive a mapping of their locations through the processing of respective geographical information. This facilitates the capture of information related to the distribution of POIs. Furthermore, preprocessing steps enable the extraction of the social activity level from the dataset.

As illustrated in Figure 3, we utilize a self-attention mechanism to capture the feature representation of exterior factor (f∈BSs,POIs,Socials). Mapping the input $M_{f}$ to distinct feature spaces, with query $Q_{f}=C_{q}M_{f}\in R^{z_{k}\times 0}$, key $K_{f}=C_{k}M_{f}\in R^{z_{k}\times 0}$, and value $V_{f}=C_{v}M_{f}\in R^{z_{v}\times 0}$, where $\left\{C_{q},C_{k},C_{v}\right\}$ represents a set of 1×1 convolutional weights, and $z_{k}$ and $z_{v}$ denote channel numbers.

Utilizing matrix multiplication to compute the similarity scores for each pair of points yields: (2)$$f=\frac{Q_{f}^{T}K_{f}}{\sqrt{z_{k}}}\in R^{O\times O}.$$

The similarity between point *i* and point *j* can be expressed as: (3)$$f_{i,j}=\frac{\left(M_{f,i}^{T}C_{q}^{T}\right)\left(C_{k}M_{f,j}\right)}{\sqrt{Z_{k}}}.$$
where $M_{f,i}$ and $M_{f,j}$ are feature scalars of shape $z_{k}\times 1$. Subsequently, the index of similarity is column-normalized: (4)$$\alpha_{i,j}=\frac{exp(f_{i,j})}{\sum_{k=1}^{N}exp\left(f_{j,k}\right)},\;i,j\in \left\{1,2,\ldots ,N\right\}.$$

The aggregated feature at position *i* is computed as the weighted sum of all positions: $X_{i}=\sum_{j=1}^{N}exp\left(\alpha \left(C_{v}M_{f,j}\right)\right)$, where $C_{v}M_{f,j}\in R^{z_{v}\times 1}$ represents the *j*th column of the value $V_{f}$. Thus, in the fully connected layer, the input $X\in R^{z_{v}\times O}$ results in the output vector $w_{f}\in R^{pO\times 1}$. Finally, the vector is reshaped into an output $W_{f}\in R^{p\times O}$. $w_{f}=\sigma \left(C_{e}X+a_{f}\right)$, $W_{f}=Resahpe\left(w_{f}\right)$, where $C_{e}$ and $a_{f}$ constitute the learnable parameters of the fully connected layer.

### 3.5. Fusion Output Module

From the above analysis, it can be discerned that the traffic data of the community are not only related to the continuous flow data but also associated with the time periods. Initially, we integrate the relevant features *Z* to capture this relationship. Subsequently, multiple DenseBlocks are employed to extract fused features, as illustrated in Figure 4. We obtain the predicted $\hat{M}$ through a sigmoid activation function.
(5)$$Z=concat\left(Y_{spatial},Y_{time},W_{f}\right).$$
(6)$$\hat{M}=\sigma \left(DenseBlock\left(Z\right)\right).$$

## 4. Experiment

### 4.1. Dataset and Experimental Settings

Performing performance prediction by collecting engineering parameter data and performance indicator data of the target radio cell using the target performance indicator prediction algorithm can help obtain a performance prediction result of the target radio cell. By selecting a site of a mobile communication operator in an area as a research object, engineering parameters of a wireless cell in the area were obtained, including the longitude and latitude of the site, site height, site type, azimuth, downtilt, power, bandwidth, and antenna model. We collected performance indicator data of cells in a region through the network management platform of a mobile carrier, including physical resource block utilization, downlink radio resource control user connection number, uplink radio resource control user connection number, and traffic volume. Employing min-max normalization, all data were scaled to the interval [0, 1], subsequently rescaled to their original values for a comparative assessment against the ground truth.

All training and testing processes were performed on an RTX 3080 (manufactured by NVIDIA in Santa Clara, CA, USA) with 10 GB of memory. The training process used the Adam optimizer. TSENet employed the widely used polynomial learning rate decay strategy to minimize the MSE loss during training. The initial learning rate was 0.01. The learning rate decayed by factors of 10 at 50% and 100 at 75% of epochs. The primary TSENet model was trained with a batch size of 16, while smaller models (batch size of 32) underwent 500 epochs. Within the convolutional module, the last layer incorporated a 1×1 filter with a sigmoid activation function, aiming to minimize the mean squared error loss during the training process. Additionally, the remaining layers were equipped with 3×3 filters comprising 16 kernels and ReLU activation functions.

The evaluation and comparison were conducted through the use of the normalized root-mean-square error (NRMSE), mean absolute error (MAE), and regression score (RS). A smaller value for the MAE and NRMSE, or a value closer to 1 for R2, signifies superior performance.

### 4.2. Inner Parameters of Transformers

We performed distinct ablative experiments, systematically varying the embedding dimension *d* and the number of heads *h* for each transformer. Table 1 presents the ultimate internal parameters of each transformer, while Section 4.4 provides comprehensive experimental outcomes for diverse input sequence combinations across transformers.

### 4.3. Analysis of Useful Temporal Features

In the TSM, we adeptly employed both hourly and daily data to extract temporal features, encompassing the recent and epochal aspects of time. Within our experimental domain, we further formulated a weekly dataset, denoted as $M_{t}\left(t\right)$, derived from the original matrix. We analyzed diverse time series to identify crucial temporal features for predicting cellular traffic on the TSM. Results in Table 2 demonstrate that combining proximity and daily periodic time features yielded the most effective outcomes. Longer cycle sequences tended to introduce disruptions to predictions, thereby reducing forecasting accuracy.

### 4.4. Analysis of Useful Spatial Features

In our model, each transformer necessitates two sequences as inputs. To enhance precision, we conducted experiments with varied combinations of input sequences. In Figure 5, the optimal inputs for each transformer are depicted. Spatial transformers leverage traffic data from pertinent grids and $M_{c}\left(t\right)$ to capture spatial information and self-information. Enclosed transformers and periodic transformers benefit from input sequences that incorporate recent and cyclical characteristics.

### 4.5. Experiment Analysis

We conducted experiments on our in-house dataset of cellular network traffic to contrast the proposed TSENet against alternative models, and the assessment outcomes are illustrated in Table 3. As shown, ARIMA exhibited an elevated MAE and RMSE on the dataset compared to alternative models, primarily because it disregarded other dependencies and concentrated solely on temporal features. SLTM’s performance was inferior to deep learning approaches but surpassed statistical methods. HSTNet exclusively attended to temporal attributes, overlooking external factors such as BS information and POI distribution. Similarly, STDenseNet neglected external factors. Although TWACNet utilizes convolution-based networks, its performance fell short of ConvLSTM-based methods. STCNet employs ConvLSTM-based networks but lacks the incorporation of self-attention and correlation layers for improved feature extraction. The aforementioned techniques heavily rely on networks for uncovering latent information, potentially inadequately modeling continuous spatiotemporal features. Compared to other methods, TSENet achieved the best performance. The TSENet model took 2765 s to complete 10 epochs during the training phase. On one side, the TSM and SSM amplify the capacity to capture continuous spatiotemporal features. Furthermore, TSENet introduces a self-attention mechanism, aggregating the capability to capture dependencies between extracted external information and network traffic information.

### 4.6. Validation of Key Components

(1)Augmentation of data: When setting Q to zero, it is equivalent to omitting data augmentation from the TSM. As discerned from Figure 5, the augmentation of data enhanced the predictive outcomes, with the minimum value of MAE observed when Q was set to 15.(2)Grid selection: Setting K to one is comparable to removing grid selection from the set-top box. As evidenced in Figure 5, grid selection markedly enhanced predictive performance. Optimal performance was achieved when selecting 20 grids.(3)Spatial features in the SSM: In order to substantiate the significance of extracting spatial features, we conducted experiments solely utilizing the TSM. The outcomes of TSENet (TSM only) presented in Table 4 underscore the efficacy of capturing spatial relationships. Additionally, we utilized a graph convolutional network (GCN) [33] in lieu of the SSM to represent spatial features in TSENet, assessing the spatial modeling efficacy of the SSM through experiments. The results in Table 4 demonstrate that incorporating the SSM yielded better performance compared to a GCN, enhancing the overall model performance.(4)Temporal fusion with the SSM: We also trained TSENet (without SSM fusion), where SSM was excluded from TSM. As illustrated in Table 4, TSENet (excluding SSM fusion) demonstrated superior performance in anomalous scenarios, confirming the spatial transformer’s effectiveness in capturing real-time spatial dependence.

**Table 4 sensors-24-01713-t004:** Transformer-based verification of key components.

Methods	MAE	NRMSE	$R^{2}$
TSENet (TSM only)	10.3499	0.54258	0.82695
TSENet (with GCN)	10.9125	0.57528	0.80473
TSENet (w/o SSM in fusion)	9.9268	0.53369	0.83329
TSENet	9.758	0.51858	0.84302

## 5. Conclusions

This paper introduced a novel TSENet to simultaneously explore spatial sequences, temporal sequences, and external factor information. TSENet treats cellular data across grids within a defined time interval as a spatial sequence, facilitating the modeling of real-time spatial correlations globally. Additionally, incorporating a self-attention mechanism, this method integrates the capability to capture dependencies among external factors and network traffic features. Experimental results showed the efficacy of TSENet, affirming the utility of the transformer architecture in cellular traffic prediction. This suggests that our proposed approach can enhance the accuracy of cellular network traffic forecasting. Future endeavors will involve developing more efficient transformers, unearthing additional valuable external information, and further refining predictions.

## Figures and Tables

**Figure 1 sensors-24-01713-f001:**
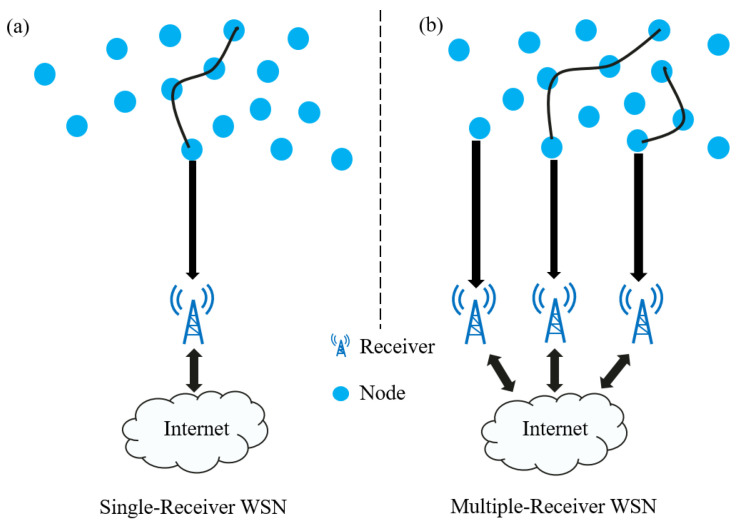
(**a**) Single-receiver WSN. (**b**) Multiple-receiver WSN.

**Figure 2 sensors-24-01713-f002:**
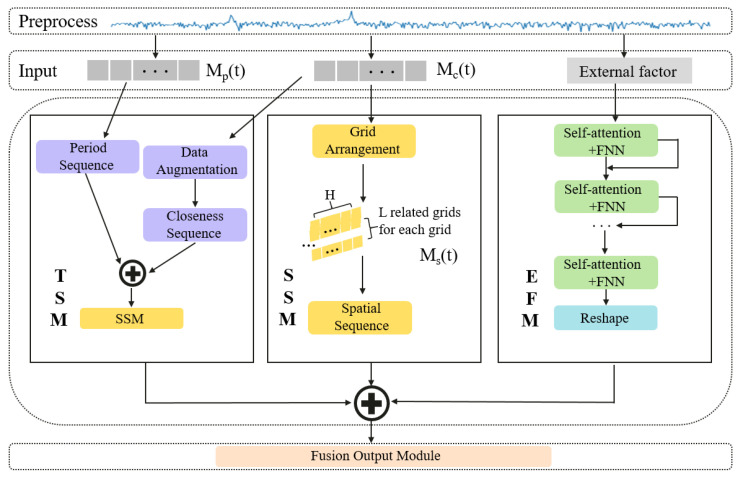
Overview of the proposed TSENet.

**Figure 3 sensors-24-01713-f003:**
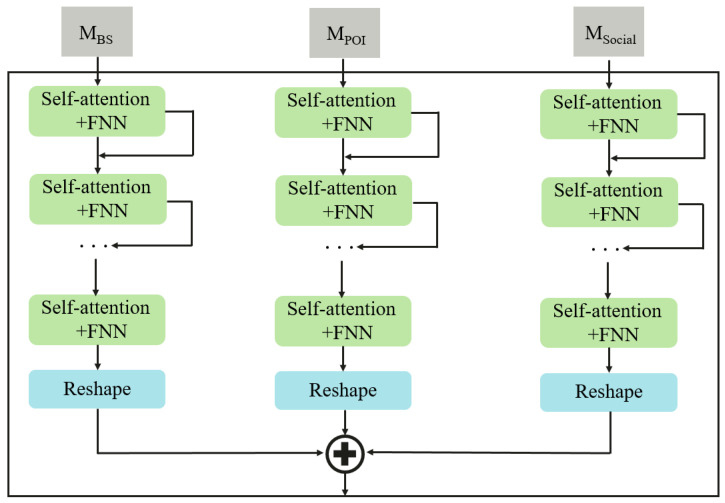
Structures of the external-factor module.

**Figure 4 sensors-24-01713-f004:**
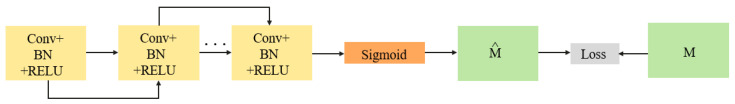
Structures of the fusion output module.

**Figure 5 sensors-24-01713-f005:**
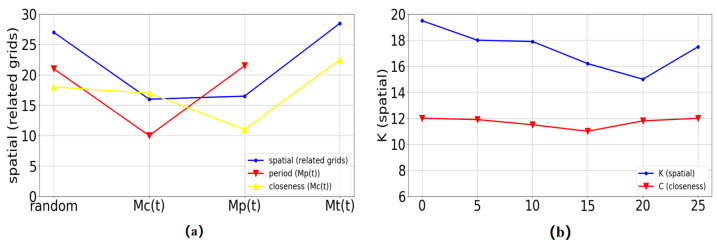
(**a**) MAE values when different transformers employed distinct initial sequences, (**b**) MAE values for SSM and enclosed transformers with varied K and Q.

**Table 1 sensors-24-01713-t001:** Parameters of different transformers.

Transformer	d	h	Source Sequence		Initial Sequence	
Spatial	64	8	Augmented	$M_{s}\left(t\right)$	Average of	$M_{c}\left(t\right)$
Closeness	256	8		$M_{c}\left(t\right)$		$M_{p}\left(t\right)$
Period	128	8		$M_{p}\left(t\right)$		$M_{c}\left(t\right)$

**Table 2 sensors-24-01713-t002:** Different combinations of time converters.

Combination	MAE	NRMSE	$R^{2}$
$M_{c}$ (hourly)	11.3955	0.55332	0.81508
$M_{p}$ (hourly)	10.5194	0.57136	0.81226
$M_{c}+M_{p}$	10.4823	0.55282	0.81539
$M_{c}+M_{t}$ (weekly)	17.6349	0.86621	0.57041
$M_{p}+M_{t}$	17.6375	0.77772	0.63955
$M_{c}+M_{p}+M_{t}$	17.4319	0.74192	0.67187

**Table 3 sensors-24-01713-t003:** Experimental results of the RMSE and MAE.

Methods	MAE	NRMSE	$R^{2}$
ARIMA	21.19	0.8364	0.3574
LSTM	15.341	0.7613	0.7275
DenseNet	13.243	0.6241	0.7830
TWACNet	11.22	0.6198	0.8021
HSTNet	10.930	0.5549	0.8295
STCNet	10.898	0.5766	0.8271
TSENet (Ours)	9.758	0.51858	0.84302

## Data Availability

Data is unavailable due to privacy reasons.

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
