# Peer review of "A TSENet Model for Predicting Cellular Network Traffic"

_sensors, 2024, doi:10.3390/s24061713_

Round 1

Reviewer 1 Report

Comments and Suggestions for Authors

1. One of my main concerns about the article is the connection between the presented method of traffic prediction and the WSN. The concept of WSN itself is not entirely up-to-date, and in addition, mobile networks are not WSNs, since they are generally not sensory. Insertions in the abstract and related work regarding WSN look irrelevant and not connected with the rest of the article. Additionally, references [1-4] also seems not very relevant and recent. 

1a. If you decideed to leave WSNs in your article, please take a closer look to Figure 1. My concern is that there here is only one receiver on multireceiver part of the figure. If this figure was inspired by Figure 1 in [14], then for my mind conception of what is receiver is quite different on these pictures.

1b. In line 97 reference [16] doesn't match the text, for my opinion. Article [16] is about clustering in WSN but not their purpose and principles of operation.

2. In lines 111-113 there is a following statement:

"While statistical models exhibit commendable performance on resource-constrained mobile devices due to their modest computational requirements" 

I have a question acording to this statement. Is there any sence to perform traffic prediction on resource-constrained mobile devices or better do it with the cloud computations, for example? If second, this argument actually doesn't make sence. Article [19] also do not use this argument. 

3. I am not sure if the conclusion in lines 118-125 (despite the fact that I personally agree with it) could be done from articles [17-22]. I'd propose to enlarge this section with more related articles. For example, a few detailed surveys on the topic has been published recently, some of them:

a. Boutaba, R., Salahuddin, M.A., Limam, N. et al. A comprehensive survey on machine learning for networking: evolution, applications and research opportunities. J Internet Serv Appl 9, 16 (2018). https://doi.org/10.1186/s13174-018-0087-2

b. Lohrasbinasab I, Shahraki A, Taherkordi A, Delia Jurcut A. From statistical- to machine learning-based network traffic prediction. Trans Emerging Tel Tech. 2022; 33(4):e4394. doi:10.1002/ett.4394

4. For my mind Figure 2. requires some improvement. For example, it is not quite clear from the picture how Mp(t) is destributed from the Input block to TSM and SSM blocks. Also is there should be Mc(t) and Ms(t) also mentioned in the picture? 

5. In section 4.3 there are three exterior factors are considered: BSs, POIs, and Socials. For my opinion more detailed information about them is needed, as well as why they were chosen out of all possible external factors.

6. Could you please provide more information about data source and dataset size and structure in the Section 4.1? Also, information about dataset preprocessing could be added to the Section. 

7. For my mind, plots in Figure 5 should be significantly improved. Axes labels either confusing, or missing, graph names are also confusing. Description of the graphs help to understand them, but they should be quite clear themselves. 

8. It would be nice to have comparison of Computational times (both training and inference) of considered models.

9. Please consider adding more graphs to the article to improve the understanding of information. For example, you can improve graphs in Figure 5, as mentioned above, or add graphs of losses dependency on other model hyperparameters, or compare a small segment of real traffic with model predictions.

Also, I have a few minor editorial comments:

1. Article [18] does't mention Holt-Winter algorithm, as it is declared in line 109. 

2. Line 115: in article [20] not ML models in general are compared to the statistical methods, but one specific ML model. For my mind it is better to name this model and do not use more general "machine learning models".

3. Line 134: ARIMA not an ML method.

4. Line 198: I suppose there should be reference to Figure 2 not Figure 1.

5. For my mind it is better to place large formulas (193, 235, 236, 238) on a separate line rather than in the text.

6. Line 326: I suppose there should be reference to Table 4, not 3.

Reviewer 2 Report

Comments and Suggestions for Authors

The authors present their implementation of a network traffic model for cellular sensor networks.  However, their paper is confused as of the differences between network sensing and environmental sensing, or it does not clearly delineate between the two.  It is not necessarily clear what "environmental" conditions the sensors are using to determine the features needed for the network traffic estimations.

The authors should describe the reasoning for their model implementations (FFM) in greater details and describe the actual inputs in detail as well.  Otherwise, the reader is left without information on why for specific data this model configuration was chosen.  Additionally, the authors should p[rovide the reasoning for the fusion output module as well.  It seems that part of the module description ended up in Section 4.1.  

The evaluation of the model using in-house generated private data for initial testing is ok, but a perfomance analysis and comparison with state-of-the-art alternatives requires that a public dataset is used to enable reproducibility.  This last point should be considered throughout the paper to positively support the authors' claims.

Comments on the Quality of English Language

The paper is in need of significant English language editing.  Similarly, close-sounding words such as "spacial" and "spatial" are used interchangeably and statements like "significant significance" are troublesome.  In parts, the paper is hard to follow as there are confusing statements or statements are not clearly worked out, which might contribute to some of the confusion on the content part.

Reviewer 3 Report

Comments and Suggestions for Authors

1.In the paper the authors proposed a novel method (TSENet framework) for simultaneously explore spatial sequences, temporal sequences, and external factor information. TSENet is based on composition of transformers and self-attention network. The method is targeted to prediction of cellular network traffic. The topic of the article is very prospective in the various studies of WSNs.

2.The authors have designed a temporal transformer module for extracting temporal features. This module accomplishes this by modeling the traffic flow within each grid of the communication network at both near-term and periodical intervals. In addition the authors amalgamate the spacial features of each grid with information from its correlated grids, generating spacial predictions within the spacial transformer.

3.In the method an integration of the the capability to capture dependencies among external factors and network traffic features is implemented.

4.The description of the suggested method (TSENet framework) is presented at good level (including related works, engineering and mathematical description, illustrative figures).

5.The Experimental results are presented at a very good level (dataset and experimental settings,

experiments results analysis). The results show the effectiveness of TSENet framework for cellular traffic prediction in sensor networks.

6.The conclusion section involves the suggestion for future research directions for the refinement of the prediction process.

7.The article can be accepted (as is).
